# Graphene balls for lithium rechargeable batteries with fast charging and high volumetric energy densities

In Hyuk Son[1], Jong Hwan Park [1,2], Seongyong Park[3], Kwangjin Park[1], Sangil Han[4], Jaeho Shin[5,6], Seok-Gwang Doo[1], Yunil Hwang[1], Hyuk Chang[1,7] & Jang Wook Choi [5,6]

Improving one property without sacrificing others is challenging for lithium-ion batteries due to the trade-off nature among key parameters. Here we report a chemical vapor deposition process to grow a graphene–silica assembly, called a graphene ball. Its hierarchical three-dimensional structure with the silicon oxide nanoparticle center allows even 1 wt% graphene ball to be uniformly coated onto a nickel-rich layered cathode via scalable Nobilta milling. The graphene-ball coating improves cycle life and fast charging capability by suppressing detrimental side reactions and providing efficient conductive pathways. The graphene ball itself also serves as an anode material with a high specific capacity of 716.2 mAh g$^{-1}$. A full-cell incorporating graphene balls increases the volumetric energy density by 27.6% compared to a control cell without graphene balls, showing the possibility of achieving 800 Wh L$^{-1}$ in a commercial cell setting, along with a high cyclability of 78.6% capacity retention after 500 cycles at 5C and 60 °C.

[1] Energy Material Lab, Material Research Center, Samsung Advanced Institute of Technology, Samsung Electronics Co., LTD, 130 Samsung-ro, Yeongtong-gu, Suwon-si, Gyeonggi-do 16678, Republic of Korea. [2] Nano Hybrid Technology Research Center, Creative and Fundamental Research Division, Korea Electrotechnology Research Institute (KERI), 12, Bulmosan-ro 10 beon-gil, Seongsan-gu, Changwon-si, Gyeongsangnam-do 51543, Republic of Korea. [3] Analytical Engineering Group, Platform Technology Lab, Samsung Advanced Institute of Technology, Samsung Electronics Co., LTD, 130 Samsung-ro, Yeongtong-gu, Suwon-si, Gyeonggi-do 16678, Republic of Korea. [4] Platform Material Team 1, SDI R&D Center, Samsung SDI Co., LTD, 130 Samsung-ro, Yeongtong-gu, Suwon-si, Gyeonggi-do 16678, Republic of Korea. [5] School of Chemical and Biological Engineering and Institute of Chemical Processes, Seoul National University, 1 Gwanak-ro, Gwanak-gu, Seoul 08826, Republic of Korea. [6] Graduate School of Energy, Environment, Water, and Sustainability (EEWS) and KAIST Institute NanoCentury, Korea Advanced Institute of Science and Technology (KAIST), 291 Daehak-ro, Yousung-gu, Daejeon 34141, Republic of Korea. [7] SDI R&D Center, Samsung SDI Co., LTD, 130 Samsung-ro, Yeongtong-gu, Suwon-si, Gyeonggi-do 16678, Republic of Korea. Correspondence and requests for materials should be addressed to I.H.S. (email: inhyuk74.son@samsung.com) or to J.W.C. (email: jangwookchoi@snu.ac.kr)

The ongoing development of lithium-ion batteries (LIBs) has resulted in remarkable improvements in various aspects of their operation, including energy density, power density, cycle life, and safety. At present, mobile IT devices account for a significant portion of LIB applications, where the operating specifications of current state-of-the-art LIBs are mostly satisfactory[1–3]. However, as electric vehicles (EVs) have penetrated LIB markets, key electrochemical properties have imposed more challenging standards; while higher energy densities are desired for increased driving mileage, enhanced reaction kinetics are demanded for fast charging and high rate operations. Safety is also definitely a critical factor for EV applications. A formidable challenge lies in overcoming the trade-off relation among these key properties; it is usually nontrivial to improve one property without sacrificing others.

Such a trade-off relation is particularly obvious between energy density and fast charging (or power capability). While the use of nanomaterials as active components[4,5] and incorporation of carbon nanomaterials[6–8] as conductive agents have demonstrated improved charging rates by reducing ion diffusion distance and internal resistance, most of those approaches still require further improvement before they can be implemented in current LIB technology. The tap-densities of nanomaterials are far lower than those of conventional micro-particles, which is detrimental to volumetric energy density, a crucial factor for many LIB applications than the gravimetric counterpart. In a similar context toward reduction in internal resistance, the integration of carbon nanomaterials such as graphene has proven effective in increasing the electric conductivity of lab-scale electrodes. However, achieving a uniform distribution of a small weight content (i.e., <3 wt%) of carbon nanomaterials in a large-scale slurry process has yet to be demonstrated. The modification of existing active materials by doping with foreign elements[9,10] or stoichiometric control[11–13] can also increase the diffusivity of Li ions and thus the charging rate of conventional active materials with micrometer dimensions. Most of those approaches are, however, effective at the expense of specific capacity.

Furthermore, beside energy density and fast charging, achieving long cycle life, especially at high temperatures (i.e., 60 °C), still remains a challenge for advanced LIBs that incorporate high-capacity electrode materials. While nickel (Ni)-rich and Li-rich layered oxide materials are considered as upcoming cathode materials due to their superior specific capacities compared to those of classical $LiCoO_2$ counterparts, it is well known that the inevitable cation-mixing during charging degrades the layered framework to spinel or rock-salt structures, along with unwanted side reactions at the electrode surface.

Considering the efforts undertaken so far, one of the most realistic solutions for immediately achieving fast charging of cathode materials with a negligible loss in energy density and cycle life involves finding conductive protective materials that can be coated uniformly on the active materials with a minimal content. Finding advanced anode materials is also essential because current graphite anodes suffer from Li metal deposition upon high rate charging.

Here we report a graphene–silica ($SiO_x$) assembly, called a graphene ball (GB), as a coating material for high-capacity Ni-rich layered cathode materials as well as an LIB anode material. Each GB is composed of a $SiO_x$ nanoparticle center and surrounding graphene layers, constituting a three-dimensional (3D) popcorn-like structure. The $SiO_x$ nanoparticles play a crucial role in multiple aspects, such as avoiding the formation of a silicon carbide (SiC) layer at the $SiO_x$–graphene interface during graphene growth, ensuring uniform coating of GB onto the cathode material, and providing a high specific capacity when GB is used as an anode material. The uniform coating of GB on the Ni-rich layered cathode enhances the interfacial stability with the electrolyte and the electronic conductivity over the electrode, improving the cyclability and fast charging capability of the cathode substantially. Taking the unique advantages of GB, the full-cell consisting of the GB-coated cathode and GB anode demonstrates the possibility of high volumetric energy density near 800 Wh $L^{-1}$ in a commercial cell condition, together with 78.6% capacity retention after 500 cycles at 5C and 60 °C.

## Results

**Synthesis of graphene ball**. In order to synthesize the popcorn-like GB, we developed a chemical vapor deposition (CVD) process for graphene growth. $CH_4$ gas was fed into a furnace in the presence of $SiO_2$ nanoparticles with diameters of 20–30 nm at 1000 °C. At this temperature, $CH_4$ is decomposed to generate hydrogen atoms, which can subsequently reduce $SiO_2$ to $SiO_x$ ($x < 2$). $OH^-$ is also simultaneously produced via the following reaction:

$$SiO_2 + CH_4 \rightarrow SiO_x + OH^- + 3H^+ + carbon(graphene) \quad (1)$$

The key feature in this reaction is that the produced $SiO_x$ provides catalytic sites for graphene growth, and $OH^-$ serves as a mild oxidant to facilitate the graphitic carbon formation toward graphene[14,15]. The role of mild oxidants on graphitic carbon formation was recently rationalized[16] in a way that, without a mild oxidant, the formation of amorphous carbon is more dominant. However, in the presence of a mild oxidant, it was discovered that the reaction toward CO or $CO_2$ formation is preferred for carbon atoms that would otherwise have evolved into amorphous carbon. In fact, the facile growth of graphene using metal catalysts is known to occur in a similar manner[17–19], where the oxygen remaining on the metal surface leads to the preferential reaction toward graphitic carbon formation. The presence of a mild oxidant prevents the formation of silicon carbide (SiC) layer at the surface of $SiO_x$ because the oxidant prevents the reduction of Si toward SiC formation[19].

In our reaction scheme based on the exclusive use of $CH_4$, at temperatures below 900 °C, graphene barely grew. In contrast, at 1050 °C and above, graphene growth became too aggressive toward graphite formation (Supplementary Fig. 1). Notably, the growth temperature of 1000 °C is lower than those (1100–1400 °C) usually employed for CVD-based graphene growth based on Si and $SiO_2$ substrates without metal catalysts[20–22]. The critical role of the growth environment was clarified by a control experiment (Supplementary Fig. 2); following the conventional CVD processes using metal catalysts, $CH_4$ and $H_2$ were fed in a 3:7 volumetric flow ratio at 1000 °C. Because the decomposition of $CH_4$ was hindered by the large quantity of $H_2$, graphene growth was indeed very slow, revealing the importance of the exclusive use of $CH_4$ on the efficient graphene growth in our synthesis protocol. Overall, the choice of reactants and reaction temperature in the present reaction scheme is distinct from previous CVD-based counterparts employing metal catalysts and plays a key role in producing a graphene–$SiO_x$ assembly with the 3D hierarchical structure.

A series of scanning electron microscope (SEM) (Supplementary Fig. 3a–f) and transmission electron microscope (TEM) analyses (Fig. 1a–g) conducted during the course of the CVD process reflect the changes in graphene growth and morphology. After 5 min of the CVD process, a 2–3-layer-thick graphene film was vaguely observed along the surface of the $SiO_2$ (Fig. 1b, e), with the powder color turning black (Supplementary Fig. 3g). The graphene growth became more prominent after 30 min of the CVD process, where 3D sheet morphology was clearly visible (Fig. 1c, f). The high-magnification TEM image (Fig. 1g) captures

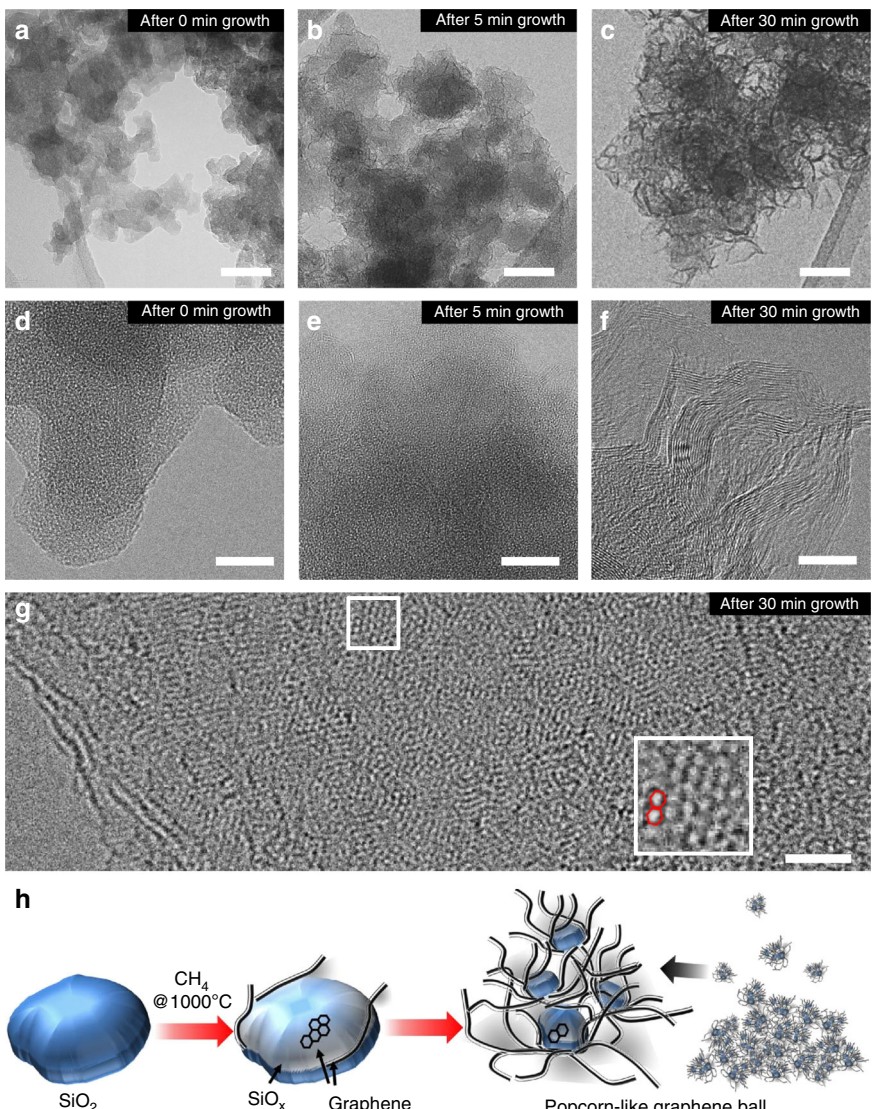

**Fig. 1** Graphene growth from $SiO_2$ nanoparticles. **a–c** TEM characterization **a** before CVD growth, **b** after 5 min growth, and **c** after 30 min growth (scale bars, 50 nm). **d–f** Their respective magnified images (scale bars, 10 nm). **g** Higher magnification image of graphene after 30 min growth and its atom-level view from the white box (inset) (scale bar, 2 nm). **h** Graphical illustration of popcorn-like graphene growth from $SiO_2$ nanoparticles

the hexagonal atomic arrangements in a section of graphene, which reflect its basal plane. After 30 min, it was also observed that the sizes of the $SiO_2$ nanoparticles decreased from 28.2 to 15.3 nm on average (Supplementary Fig. 4). This occurred because graphene growth involves the aforementioned reduction of $SiO_2$ and subsequent $OH^-$ formation. Hence, the overall graphene growth is fairly analogous to popcorn expanding in the 3D bulky structure, produced at the expense of the original kernels (Fig. 1h).

The graphene growth was further elucidated by a series of additional analyses. X-ray diffraction (XRD) spectra obtained during the growth indicate that as the CVD process progresses, the peaks at $2\theta = 26.44°$ and $44°$, corresponding to the (002) and (101) planes of graphite, grew continuously (Fig. 2a), supporting once again the formation of graphitic carbon[23–25]. The graphene growth was also reflected in the peaks at 284.5 and 291.5 eV in the C 1s branch of X-ray photoelectron spectroscopy (XPS) analysis (Fig. 2b)[22,23]. The peak at 284.5 eV is assigned to C–C and C = C bonds, which appeared prominently after 30 min of growth due to graphitic carbon formation. The progressive XPS peaks in the Si 1s branch of XPS analysis (Fig. 2c) reconfirm[25] the reduction of

$SiO_2$ during the CVD process. With increasing growth time, the peak near 103 eV shifted to lower binding energies due to the decreased oxidation state of Si. Based on the presence and intensity of the carbon signal in energy dispersive X-ray (EDX) mapping (Fig. 2d–g), it was found that even after 5 min, graphene growth indeed took place along the surface of the $SiO_x$, and after 30 min, the growth spread over the entire image area, while Si and O remained homogeneously distributed. The weight portion of graphene, and the results of Raman analysis, specific surface area, and conductivity of GB at different CVD growth time are summarized in Table 1. As for the specific surface area, its decrease with increasing reaction time is attributed to the progressive graphitization during which graphene layers are merged and stacked. Detailed Raman and thermogravimetric (TG) analyses spectra are presented in Supplementary Fig. 5.

**Coating of graphene ball onto Ni-rich cathode material.** Because of its atom-thick-layered morphology, graphene has been considered to be one of the most ideal materials for surface protective coating of high capacity layered cathode materials.

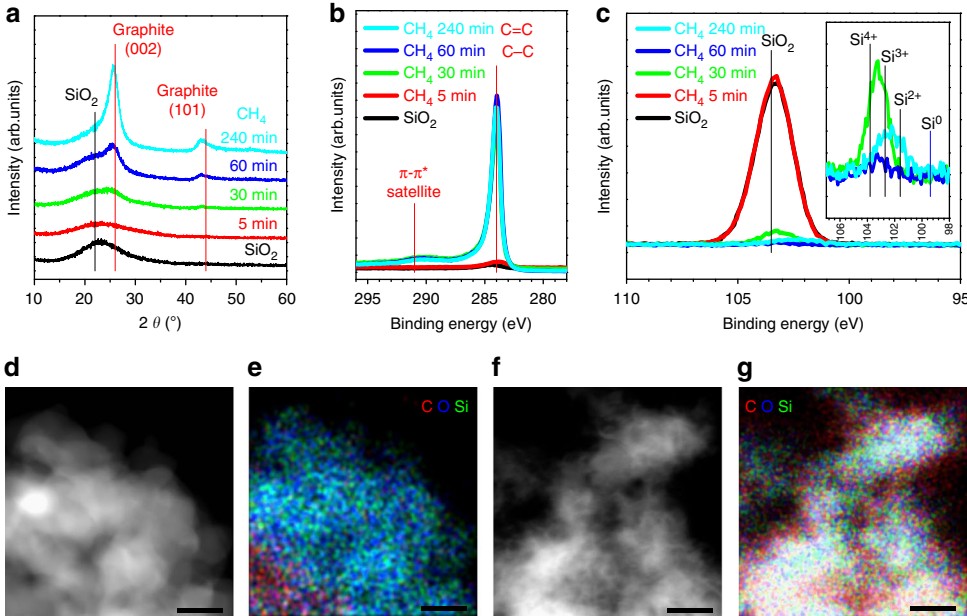

**Fig. 2** Analysis of graphene-ball during its growth. **a–c** Monitoring graphene growth during the course of the CVD process using **a** XRD, **b** XPS in C 1s branch, and **c** XPS Si 2p branch. **d–g** EDX elemental mapping with respect to C, O, and Si after **d, e** 5 min and **f, g** 30 min graphene growth from SiO$_2$ (scale bars, 50 nm)

**Table 1 Various properties of graphene ball at different CVD reaction time**

| Time (min) | Graphene content in mass (%) | D/G peak ratio | Specific surface area (m$^2$ g$^{-1}$) | Total pore volume (cm$^3$ g$^{-1}$) | Conductivity (S cm$^{-1}$) |
|---|---|---|---|---|---|
| 0 | — | — | 175.5 | 0.5821 | < 10$^{-7}$ |
| 5 | 17.1 | 1.102 | 147.1 | 0.5535 | 5.8 |
| 30 | 42.5 | 1.076 | 108.3 | 0.2875 | 22.6 |
| 60 | 56.2 | 1.066 | 81.9 | 0.2339 | 48.4 |
| 240 | 84.5 | 1.055 | 35.6 | 0.1978 | 65.5 |

However, in the case of well-known solution processes[26–29] based on Hummer's method, achieving a uniform coating with a limited content of graphene is nontrivial due to agglomeration, which arises from the strong electrostatic interaction between graphene sheets. In an effort to overcome the agglomeration issue, a different approach, such as the crushing of graphite employing a ball-milling process, has been introduced[30]. However, this process is usually slow (>6 h) and, more critically, can harm the crystallinity of the host material. It should also be noted that amorphous carbon coating by CVD processes is not applicable to Ni-rich oxide materials[31,32] because the CVD processes extract transition metals (TMs), perturbing the original crystal structure of the host material.

In contrast, the synthesized GB in this study can be coated on the surface of commercial LiNi$_{0.6}$Co$_{0.1}$Mn$_{0.3}$O$_2$ powder (size = ~10 μm) at a minimal content of 1 wt% via a mild Nobilta milling process. The coating process is also quite fast (~10 min at 3000 rpm). The SiO$_x$ seeds play a critical role in enabling this uniform coating at low weight contents. The SiO$_x$ seeds account for 57.5 wt% of the GB (TG result in Supplementary Fig. 5b) and in the coating process, their presence allows the GB to exert a shear force onto the surface of the LiNi$_{0.6}$Co$_{0.1}$Mn$_{0.3}$O$_2$, producing intimate interaction and even penetration. To confirm this phenomenon, the same coating process was undertaken with a physical mixture of 0.5 wt% SiO$_2$ and 0.5 wt% graphene. This sample suffered from crystalline damage on the surface region of the NCM due to aggressive collisions between

LiNi$_{0.6}$Co$_{0.1}$Mn$_{0.3}$O$_2$ particles as well as between LiNi$_{0.6}$Co$_{0.1}$Mn$_{0.3}$O$_2$ particles and the reactor wall (Supplementary Fig. 6). These phenomena were amplified by graphene agglomeration that occurred in the absence of the 3D hierarchical structures containing SiO$_x$ particle centers.

The uniform coating of GB was confirmed by SEM and TEM analyses. Hereafter, LiNi$_{0.6}$Co$_{0.1}$Mn$_{0.3}$O$_2$ before and after GB coating is denoted as pristine NCM and GB-NCM, respectively. Low magnification SEM images of the NCM before and after GB coating (Fig. 3a, b, Supplementary Fig. 6a, b) showed very similar morphologies because of the uniform distribution of the low content GB. However, a higher-magnification SEM image of GB-NCM indicates that GB penetrated into the spaces between the primary particles of NCM (Fig. 3d). The penetration of GB was more clearly observed in TEM images (Fig. 3e, f). In Fig. 3e, a graphene sheet was observed in the open space above an NCM particle (see red dotted lines). Some graphene sheets were aligned along the surface of the NCM and displayed a consistent interlayer distance of 3.3 or 3.4 Å (Fig. 3f).

EDX elemental mapping before and after the GB coating (Fig. 3g, h) also indicates the homogeneous distribution of GB over the entire area of each NCM particle. In Raman analysis (Fig. 3i), pristine NCM did not exhibit any peaks, whereas the coating of GB in the case of GB-NCM was reflected in its D, G, and 2D peaks, which were commensurate with those of bare GB. Similarly, graphitic carbon formation represented by C–C and C = C bonds was detected in the XPS C 1s spectrum of the

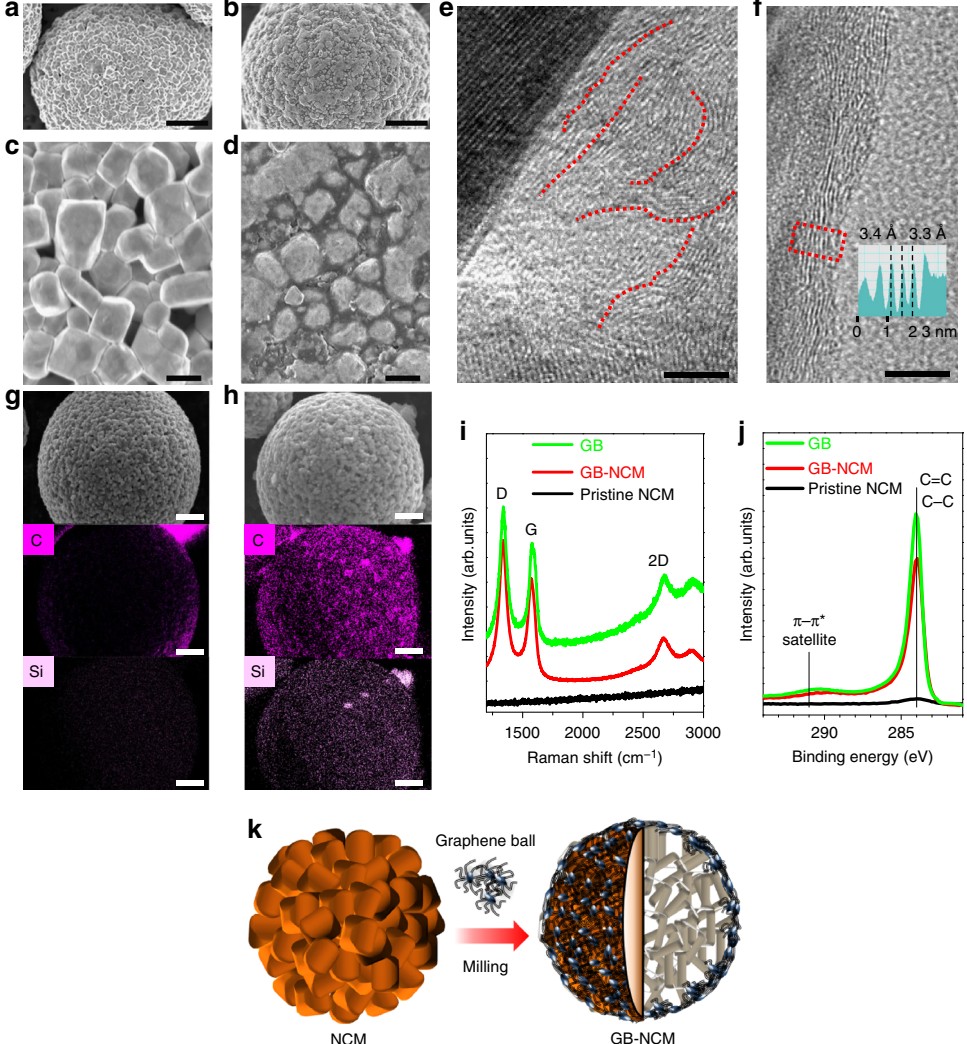

**Fig. 3** Coating of GB onto LiNi$_{0.6}$Co$_{0.1}$Mn$_{0.3}$O$_2$. SEM images of LiNi$_{0.6}$Co$_{0.1}$Mn$_{0.3}$O$_2$ **a**, **c** before (scale bars, 2 μm and 200 nm, respectively) and **b**, **d** after GB coating (scale bars, 2 μm and 200 nm, respectively). **e**, **f** TEM images of graphene sheets (scale bars, 5 nm). **e** Located between primary particles of LiNi$_{0.6}$Co$_{0.1}$Mn$_{0.3}$O$_2$ and **f** coated along primary particles of LiNi$_{0.6}$Co$_{0.1}$Mn$_{0.3}$O$_2$. **g**, **h** EDX elemental mapping of LiNi$_{0.6}$Co$_{0.1}$Mn$_{0.3}$O$_2$ with respect to C and Si (scale bars, 2 μm). **g** Before and **h** after GB coating. **i** Raman spectra of GB, pristine LiNi$_{0.6}$Co$_{0.1}$Mn$_{0.3}$O$_2$ and GB-coated LiNi$_{0.6}$Co$_{0.1}$Mn$_{0.3}$O$_2$. **j** XPS spectra of the same three samples in C 1s branch. **k** Schematic illustration of the GB coating onto LiNi$_{0.6}$Co$_{0.1}$Mn$_{0.3}$O$_2$

GB-NCM (Fig. 3j). A satellite peak at 291.5 eV also supports graphitic carbon formation[23]. Importantly, even 1 wt% addition of GB improved the conductivity of NCM powder significantly when tested in a pelletized form; pelletized pristine NCM and GB-NCM exhibited conductivity values of $1.13 \times 10^{-3}$ and $9.72 \times 10^{-2}$ s cm$^{-1}$, respectively. Overall, the unique hierarchical structure of GB enables its uniform coating over micrometer-size NCM powder at the minimal content (Fig. 3k) and supplements the relatively low conductivity of NCM.

**Battery testing**. To test the effect of the GB coating on battery performance, both pristine NCM and GB-NCM were first examined in a half-cell configuration, using a Li metal disk as a counter and reference electrode. We first discuss the electrode conditions that we employed, to emphasize the commercial viability of the GB-based approach. In current commercial LIB cathodes based on LiCoO$_2$ (for example, those used for smart phones), the active material occupies 96–97 wt% of the electrode to maximize the volumetric energy density[3]. In the case of Ni-rich active materials, because of their lower electronic conductivity, a

content of ~92 wt% is presently accepted as an optimal value, with a conductive agent and binder each accounting for ~4 wt%.

In our case, in an attempt to compensate the relatively low conductivity of Ni-rich cathodes and thus increase the content of active material to the levels of LiCoO$_2$-based electrodes, an electrode consisting of 97 wt% NCM active material, 0.5 wt% super P, 1.0 wt% GB, and 1.5 wt% polymeric binder (polyvinylidene fluoride, PVDF) was prepared (see details in Supplementary Table 1). Since SiO$_x$ seeds constitute 57.5 wt% of the GB (Supplementary Fig. 5b), graphene itself occupies only 0.425 wt% of the entire electrode.

Remarkably, the increase in NCM content (92 wt% → 97 wt%) enabled by the enhanced conductivity due to the integrated GB increased the volumetric capacity of the electrode by ~33.3%; at the given gravimetric capacity of 190 mAh g$^{-1}$, the GB-NCM (97 wt% active material) and pristine NCM (92 wt% active material) electrodes delivered 760 and 570 mAh cc$^{-1}$, respectively (Supplementary Table 2). For reference, the tap densities of NCM in both electrodes were 4.0 and 3.0 g cc$^{-1}$, respectively. This large difference originates from the low density of super P; even a small change in the super P content leads to a significant difference in

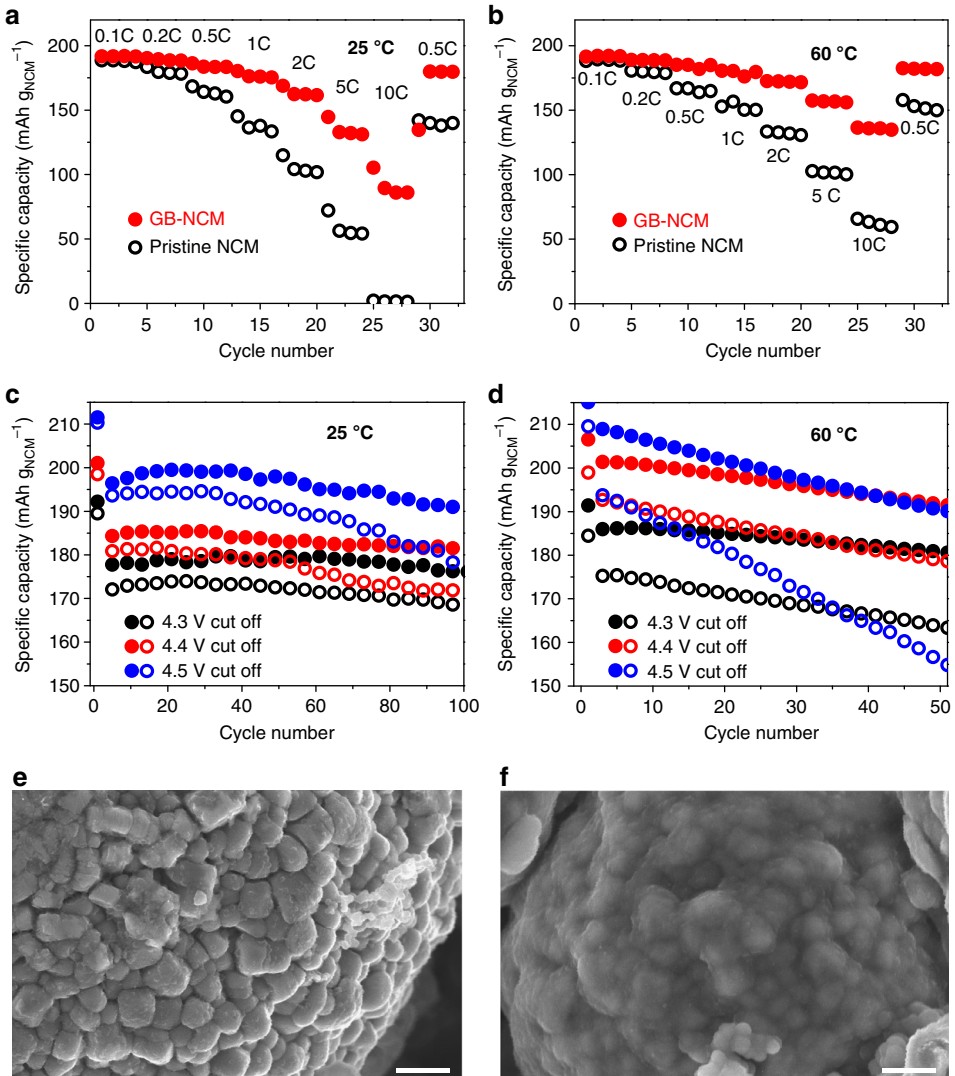

**Fig. 4** The effect of graphene-ball coating on fast charging capability and cycling stability. **a**, **b** Charging capacities of GB-NCM and pristine NCM at **a** 25 °C and **b** 60 °C when measured at different charging C-rates (1C = 190 mA g$^{-1}$) in the range of 2.5–4.3 V. The discharging rate was fixed to 1C in both **a** and **b**. **c**, **d** Discharging capacity retentions of both electrodes with different upper cutoff voltages at **c** 25 °C and **d** 60 °C. Filled: GB-NCM, open: pristine NCM. **e**, **f** SEM images (scale bars, 5 nm) of **e** pristine NCM and **f** GB-NCM after 100 cycles measured at 60 °C with 4.3 V cutoff. The mass loadings of all the electrodes (active material, binder, conductive carbon) in this figure were 25.0 ± 0.2 mg cm$^{-2}$

the tap density of the active material. In fact, the increased volumetric capacity on the cathode side has a drastic effect on the full-cell energy density, as will be noted when the GB anode is described.

As displayed in Supplementary Fig. 7a, during the first cycles of both electrodes at 0.1C (1C = 190 mA g$^{-1}$), they exhibited quite similar charging and discharging capacities (195.18 and 188.50 mAh g$^{-1}$ for pristine NCM vs. 198.24 and 191.7 mAh g$^{-1}$ for GB-NCM) as well as consistent profiles, implying that the implementation of GB did not perturb the original electrochemical properties of the NCM active phase.

However, fast charging capability was clearly different for both electrodes (Fig. 4a, b). For this testing, the charging rate was varied from 0.1 to 10C while the discharging rate was fixed at 1C, because the discharging at 1C is slow enough to discharge the full capacity regardless of charging rate. As the charging rate was raised from 0.1C to 0.2, 0.5, 1, 2, 5, and 10C, the charging capacity of GB-NCM decreased from 191.6 mAh g$^{-1}$ to 188.4, 183.5, 175.3, 161.6, 131.2, and 86.1 mAh g$^{-1}$, respectively (Fig. 4a). In contrast, with the same increases in C-rate, the charging capacity of pristine

NCM dropped more severely, from 187.2 mAh g$^{-1}$ to 178.3, 160.5, 133.4, 101.8, 54.2, and 1.3 mAh g$^{-1}$, respectively. When tested at 60 °C (Fig. 4b), the gap between both electrodes became more narrow, presumably due to the increased electronic conductivity of the NCM. However, the superior charging capability of GB-NCM was preserved; for example, while both electrodes exhibited almost the same charging capacities at 0.1C, GB-NCM showed 31.4% (171.7 vs. 130.7 mAh g$^{-1}$) and 126.7% (134.9 vs. 59.5 mAh g$^{-1}$) higher capacities at 2 and 10C, respectively.

The integration of GB also had a remarkable effect on the cyclability of NCM. It is well known that the charging process degrades the cyclability of Ni-rich layered materials; during charge, TM cations tend to migrate to the neighboring Li-slabs, leading to the so-called Li-TM mixing and consequently phase transition from the layered to a spinel even to a rock-salt phase[33]. This cation mixing accelerates the extraction of TM ions to the electrolyte, boosting side reactions at the electrode-electrolyte interface. This unstable interface is indeed fatal to the cycle life of NCM and is amplified during over-charging and at high temperature.

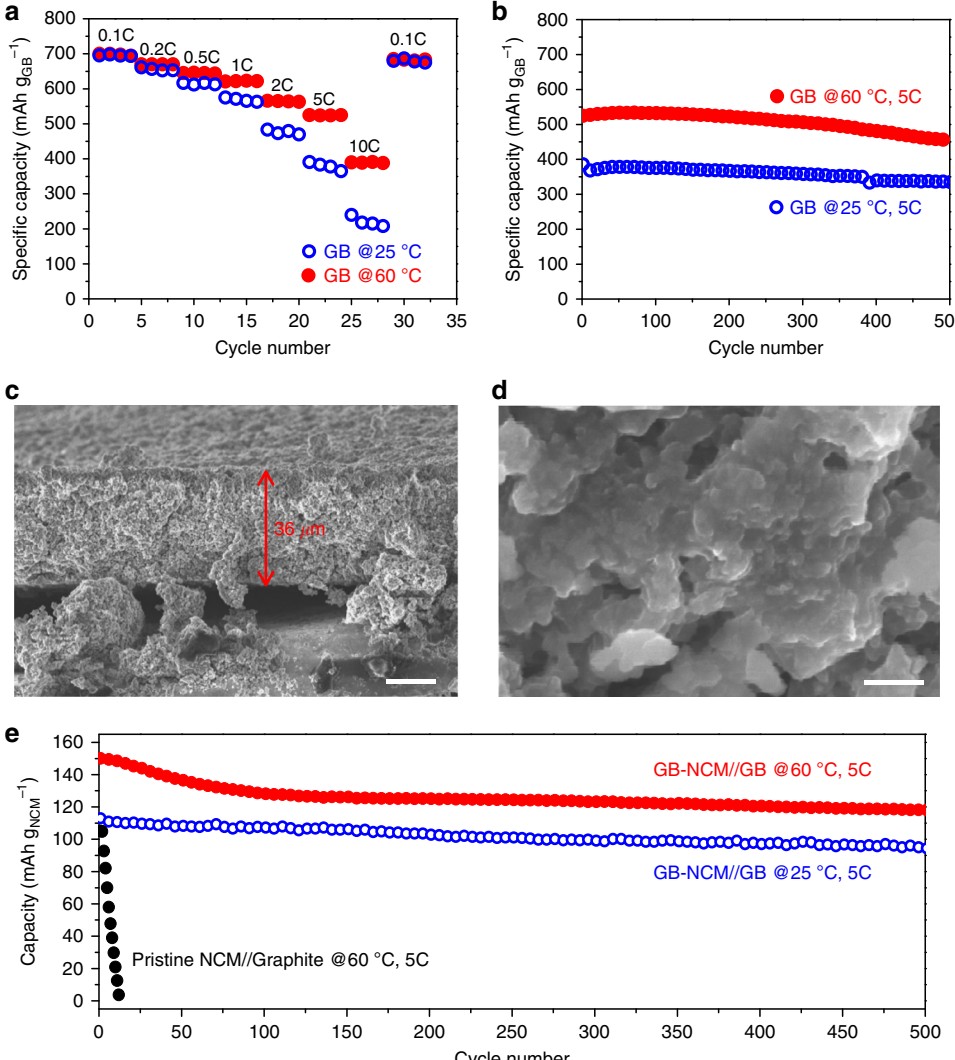

**Fig. 5** The electrochemical tests of the GB anode in half-cell and full-cell configurations. **a** Rate capability of the GB anode in half-cell when measured at various C-rates (1C = 700 mA g$^{-1}$) at 25 and 60 °C. **b** Cycle life of the GB anode at 25 and 60 °C when measured at 5C in half-cell. **c** Cross-sectional SEM image of the GB anode after 500 cycle at 5C (scale bar, 15 μm). **d** A magnified view of **c** (scale bar, 200 nm). **e** Cycle life of the GB-NCM/GB full-cell at 25 and 60 °C when measured at 5C. The initial areal capacities of the GB anodes in half-cell and full-cell are 2.7 and 2.4 mAh cm$^{-2}$, respectively. In each cycle, charging and discharging rates were the same for all the data shown in this figure

As shown in Fig. 4c, at 25 °C, GB-NCM exhibited more robust capacity retention at all of the different upper cutoff conditions from 4.3 to 4.5 V. For example, when tested for 100 cycles at 1C with a 4.5 V cutoff, GB-NCM and pristine NCM preserved 97.3% and 92.1% of the capacities in the second cycles, respectively. The first cycles with all cutoff voltages were measured at 0.1C. This superior cyclability was also retained at 60 °C (Fig. 4d), which is a critical temperature for EV operations. With the same 4.5 V cutoff, after 50 cycles, the capacity retentions of GB-NCM and pristine NCM were changed to 91.0% and 79.9% with respect to their capacities in the second cycles, respectively. The superior cycling performance of GB-NCM can also be attributed to the SiO$_x$ nanoparticles, which scavenge fatal HF[34], in addition to the surface protection of GB against side reactions involving the electrolyte. The protective role of GB was reflected in the electrochemical impedance spectroscopy (EIS) analysis (Supplementary Fig. 7b, c). Detailed specific capacities, capacity retentions, and Coulombic efficiencies of both electrodes are summarized in Supplementary Table 3.

As displayed in Fig. 4e and Supplementary Fig. 8a, the uncontrolled surface reactions of pristine NCM are reflected in

the round shape of its primary particles, compared to their more angular morphology before cycling (Fig. 3c). In contrast, the surface of GB-NCM appears to be covered with SEI layers (Fig. 4f and Supplementary Fig. 8b) because of the GB protection, which induces electrolyte decomposition preferentially on the outer surface, rather than in the inner regions. A control experiment that tested the solubility of TM to the electrolyte solvents verified the protective function of GB (Supplementary Table 4).

The GB itself can serve as an anode material in pairing with GB-NCM to support fast charging operations. As addressed above, classical graphite anodes are not usually compatible with high C-rate charging (i.e., >2C) due to the deposition of Li metal on the electrode surface. In an attempt to overcome this limitation of graphite, the GB grown for 30 min was evaluated as an anode material under the half-cell configuration. To this end, the GB anode was pressed to increase its tap density to 1.1 g cm$^{-3}$ using a cold isostatic pressing (CIP) process. The GB electrode upon the CIP process delivered a reversible capacity of 716.2 mAh g$^{-1}$ or 787.8 mAh cm$^{-3}$ at an areal capacity of 2.7 mAh cm$^{-2}$ (Supplementary Fig. 9). The organized hierarchical structure of GB turned out to be critical, as a simple mixture of

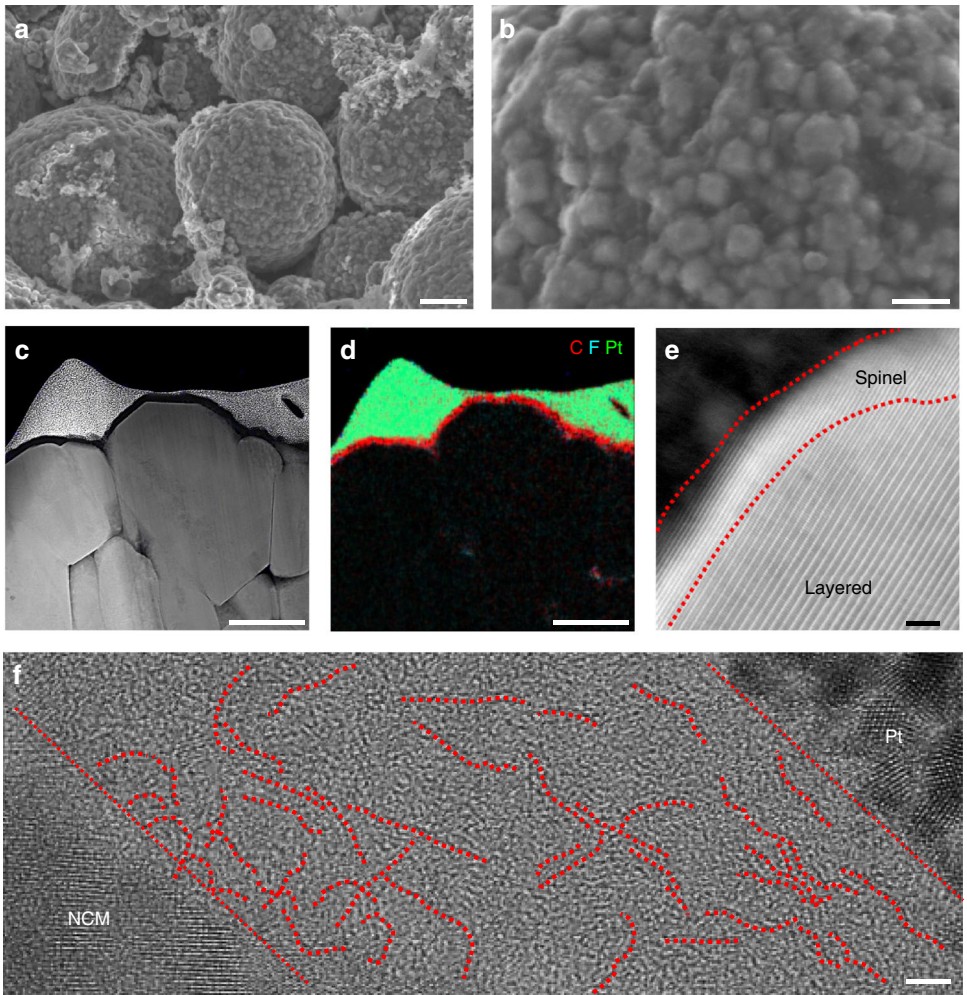

**Fig. 6** Characterization of GB-NCM in a full-cell cycled at 60 °C and 5C for 500 cycles. **a**, **b** SEM images **a** low magnifications and **b** high magnifications (scale bars, 2 μm and 500 nm, respectively). **c** Cross-sectional STEM image (scale bar, 200 nm) and **d** its EDX elemental mapping with respect to C, F, and Pt (scale bar, 200 nm). **e** High-resolution STEM image around the surface of GB-NCM (scale bar, 2 nm). **f** TEM image around the NCM surface (scale bar, 2 nm)

$SiO_2$ nanoparticles and graphene with the same compositional ratio and areal loading exhibited a specific capacity of only ~20 mAh g$^{-1}$. This inferior result reveals the importance of homogeneously distributed $SiO_x$ around graphene in achieving the electrochemical performance. At such high areal loading of GB, the GB electrode showed good rate capability such that even at 10C, the GB electrode delivered 240.6 mAh g$^{-1}$ at 25 °C, corresponding to 33.6% capacity retention with respect to the capacity measured 0.1C (Fig. 5a). The rate performance improved at 60 °C in such a way that 390.5 mAh g$^{-1}$ was observed at 10C, presumably due to the enhanced kinetics in the Li-ion reaction with $SiO_x$ particles.

The GB electrode also demonstrated robust cycling performance (Fig. 5b). At 25 °C and 5C (1C = 700 mA g$^{-1}$), starting from 382.6 mAh g$^{-1}$, the specific capacity ended at 336.7 mAh g$^{-1}$ after 500 cycles, corresponding to 88.0% retention. At 60 °C and 5 C, the specific capacity started at a higher value of 524.1 mAh g$^{-1}$ owing to the aforementioned enhanced kinetics and preserved 451.2 mAh g$^{-1}$ for 500 cycles, leading to 86.1% retention. The relative worse capacity retention at 60 °C is ascribed to less stable SEI formation around the GB, although further investigation is required to clarify. During the given 500 cycles, the electrode thickness increased only 18% from 30.5 to 36.0 μm (Fig. 5c), indicating the capability of the GB electrode to accommodate the volume change while the large specific capacity is achieved. A higher magnification SEM image (Fig. 5d) showed a well-preserved morphology of the GB with SEI layer grown on its surface. More detailed electrochemical results of the GB half-cell are presented in Supplementary Table 5.

The stable cycling of the GB half-cell prompted us to examine a full-cell in pairing with GB-NCM (Supplementary Fig. 10). For this testing, pristine NCM//graphite full-cell was also tested as a control sample. In an effort to overcome the low initial Coulombic efficiency (ICE) of the GB anode, a pre-lithiation process was adopted following the previous report[35]. The n/p ratio, defined by the capacity ratio between pre-lithiated anode and pristine cathode, was 1.0. In pairing these two electrodes in a full-cell, the following specifications of both electrodes were taken into consideration: whereas the specific capacities of the pre-lithiated GB and NCM are 716.2 and 190 mAh g$^{-1}$ at 0.1C, their ICEs are 99.8% and 93.7%, respectively. With this full-cell design, the first specific capacities of the anode and cathode in the actual cell were 666.6 and 178.2 mAh g$^{-1}$ at 25 °C, respectively, leading to an ICE of 93.1% (Supplementary Fig. 10a and Supplementary Table 6). At both 25 and 60 °C, when cycled in the voltage range of 2.8–4.2 V at 5C, the GB-NCM//GB cell exhibited excellent cyclability, such as 84.1 and 78.6% retentions after 500 cycles (Fig. 5e). The cycling performance at 60 °C is particularly

remarkable, as the pristine NCM//graphite cell decayed immediately from the beginning of the cycles. This inferior cyclability of the pristine NCM//graphite cell once again originates from Li metal deposition[36,37] on the graphite surface during high C-rate charging, in addition to the structural destabilization of pristine NCM. In this sense, the hierarchical structure of GB is suitable for fast charging and safe operation of LIB anodes by accommodating large Li-ion flux even at a commercial level of electrode loading. These series of results, in turn, reveal the difficulty of fast charging with current graphite anodes. Detailed electrochemical results of the GB-NCM//GB full-cell are presented in Supplementary Table 6. Importantly, if the GB-NCM//GB cell is applied to an EV prismatic cell that is presently available in the commercial market, the total capacity at the given cell frame would be increased by 27.6%, from 49.67 to 63.40 Ah, at 0.33 C (see details in Supplementary Table 1). Since some products consisting of a similar anode and cathode with the control cell in Supplementary Table 1 were known to deliver ~630 Wh L$^{-1}$ (http://www.greencarcongress.com/2009/12/panasonic-20091225.html), the given capacity increase predicts that the volumetric energy density of the GB-NCM//GB cell could approach 800 Wh L$^{-1}$ in a commercial cell setting, a remarkable milestone in the development of LIBs for next-generation EVs. Graphene flakes produced by the solution-based exfoliation of graphite was previously integrated with LiFePO$_4$ cathode to build a full-cell[38]. Despite the decent cyclability, the graphene nanoflake anode is expected to offer a smaller volumetric capacity than that of the GB containing high capacity SiO$_x$, leading to an inferior volumetric energy density of the corresponding full-cell.

The superior performance of the GB-NCM//GB full-cell also reconfirms the critical role of GB protection on the NCM. It is emphasized that the GB-NCM//GB cell adopts commercial electrode conditions in such a way that both the anode and cathode deliver areal capacities of 2.4 mAh cm$^{-2}$, respectively. It is also reminded[39,40] that an NCM electrode with a commercial level of volumetric capacity does not usually operate at above 2C because of its relatively low electric conductivity. In this regard, the integration of 1 wt% of GB to the NCM cathode has a dramatic effect on the cycling performance and fast charging capability even in such challenging commercial cell conditions. The effect of GB was also demonstrated for full-cells incorporating graphite anodes (Supplementary Fig. 11). Consistent with the half-cell results, the superior performance of GB-NCM//graphite full-cell as compared with that of pristine NCM//graphite counterpart is attributed to the GB's protection of NCM against side reactions and gas evolution.

The stable structural and interfacial properties of GB-NCM were further revealed by post-mortem analyses of the corresponding full-cells after 500 cycles at 60 °C (Fig. 6). The SEM images (Fig. 6a, b) demonstrate that the morphology of the NCM is consistent with that (Fig. 4f) of the half-cell after 100 cycles, implying that the beneficial effect of the GB coating is retained in full-cells over more prolonged cycles.

The cross-sectional scanning transmission electron microscope (STEM) images (Fig. 6c and Supplementary Fig. 12) show the fully filled internal morphology of GB-NCM, which indicates that the GB coating blocks electrolyte penetration and the ensuing formation of cracks and voids, a well-known failure mechanism of NCM during long-term cycling. This phenomenon is reflected by a greatly weakened fluorine signal, a signature of electrolyte penetration and decomposition, inside the GB-NCM host (Fig. 6d and Supplementary Fig. 12d), in contrast with pristine NCM, where cracks are clearly visible in both primary and secondary particle levels (Supplementary Fig. 13a–c). While a phase transition to a spinel-like structure in the 5 nm surface region of the GB-NCM was observed by STEM analysis (Fig. 6e), the

boundaries of graphene sheets were still observed (dotted lines in the TEM image of Fig. 6f) even after such aggressive cycling.

## Discussion

Fast charging capability is considered critical to the successful adoption of all-electric vehicles by the public; however, sacrificing energy density to achieve an enhanced charging rate is not desirable. To resolve this dilemma, one feasible approach for high capacity metal oxide cathodes would be to develop a uniform coating that utilizes a minimal content of a conductive carbon component to enhance electric conductivity, which could be adopted immediately with minimal commercial interruption. At the same time, advanced anode materials that can withstand high rate charging without Li metal deposition are another "must-item" to incorporate.

In this respect, the proposed addition of a limited amount of GB to NCM cathodes would not require a substantial change in the mixing conditions of slurry preparation, which is presently a major obstacle to the implementation of a new material in LIB manufacturing. Unlike many other LIB electrodes, which can be carbon-coated via an established CVD process, Ni-rich layered cathode materials are not compatible with high-temperature carbonization processes because of undesired Ni extraction, thus limiting viable options to the physical mixing of carbon nano-materials. At the same time, using physical mixing processes to produce a homogeneous electrode coating with a minimal content of carbon conductive materials is also problematic, because spontaneous agglomeration preferentially occurs, reducing interactions with the active particles.

We resolved this challenge by employing a scalable Nobilta milling process that allows GB to be homogeneously integrated with the active powder by utilizing the shear force that originates with the SiO$_x$ centers of the GB. The GB can also serve as a high-capacity anode with robust structural stability for prolonged cycles. The formation of GB is accomplished in our CVD process under exclusive CH$_4$ flow, which allows 3D graphene growth from the catalytically active sites of SiO$_x$ nanoparticles. This process serves as the basis for the enhanced charging capability and cyclability observed in this study. The current GB coating can also be applied to a range of battery electrodes that suffer from unstable interfacial properties and poor electronic conductivity.

In a broader perspective, the present mixing approach can be more generally expanded to a variety of graphene-based composites whenever it is desirable to homogeneously integrate graphene with ceramic materials. Other high-strength and heat-radiation applications could immediately benefit from the given approach.

## Methods

**CVD process for graphene-ball growth**. For the direct growth of graphene over the surface of SiO$_2$, fumed SiO$_2$ nanoparticles (average diameter: 20–30 nm, Alfa Aesar) were introduced into a fixed-bed vertical tube reactor made of quartz[18,19,31,40] (inner diameter = 27 mm, length = 80 cm). The feed gas was fed through the reactor-containing bed from the top at atmospheric pressure. The loading of SiO$_2$ in each batch was 0.3 g. Initially, the system was heated to a target temperature (900, 1000, and 1050 °C) at a ramping rate of 23 °C min$^{-1}$ while 50 sccm CH$_4$ was flowed. The temperature was maintained for 5, 30, 60, and 240 min, and the system was then cooled down to room temperature (25 °C) while pure N$_2$ gas was flowed. For the samples treated in H$_2$ atmosphere, a mixture of H$_2$ (70%) and CH$_4$ (30%) was flowed at 1000 °C for 10 min.

**Coating of graphene ball onto LiNi$_{0.6}$Co$_{0.1}$Mn$_{0.3}$O$_2$**. LiNi$_{0.6}$Co$_{0.1}$Mn$_{0.3}$O$_2$ (NCM-613, Daejung Chemicals and Materials Co. Ltd.) was first mixed with 30 min grown GB using a pestle and mortar. The GB content was 1 wt% of the entire mixture. The coating of GB was achieved at ambient temperature using a Nobilta mini bowl (Hoso-Kawa, Japan), in which a rotating four-way blade is placed with the rotation axis oriented along the reactor tube, and the gap between the blade and reactor wall is 5 mm. During the milling process, the blade was rotated at 3000 rpm for 10 min,

and as a result, a controlled shear stress was continuously applied to the GB and NCM mixture.

**Physical and chemical analyses**. X-ray diffraction (XRD) patterns were obtained using an X-ray diffractometer (D8 Advance, Bruker Inc., 40 kV, 40 mA) engaging nickel-filtered Cu Kα radiation ($\lambda = 1.54$Å). The scan rate and interval were $4° \text{min}^{-1}$ and $0.02°$, respectively. XPS measurements were performed using a Physical Electronics spectrometer (Quantera II, ULVAC-PHI, Inc.) with an Al Kα source (1486.7 eV). The carbon content of GB was quantitatively analyzed using a thermogravimetry analyzer (METTLER TOLEDO TGA/DSC1). High-resolution transmission electron microscopy (HRTEM) imaging and diffraction pattern (DP) analysis were carried out using an FEI Titan Cubed 60–300 equipped with double Cs correctors and Gatan Quantum 965. The samples were also visualized using field emission scanning electron microscopy (FE-SEM, Nova NanoSEM 450s, FEI) to monitor the changes in morphology.

The electronic conductivity of the pellets was measured using the two-probe DC method after pelletizing the active powder at 20 kN (MCP-PD511, Mitsubishi). Raman spectra were obtained using a Renishaw InVia micro-Raman spectrometer at ambient condition. A 514 nm wavelength Ar-ion laser was used as the excitation source and the incident power was limited to 2 mW to minimize the damage of graphene. The spectral resolution was $\sim 1 \text{cm}^{-1}$. Peak analyses related to intensity, position, and deconvolution were carried out using the WiRE 3.3 software. The metal ion dissolution test was performed using inductively coupled plasma-atomic emission spectroscopy (ICP-AES, IPS-8100, Shimadzu). For testing of transition metal dissolution, 0.1 g of the active powder were immersed in 20 mL of the electrolyte co-solvents (ethylene carbonate (EC):ethyl methyl carbonate (EMC): dimethyl carbonate (DEC) = 3:4:3 = v:v:v) at 50 °C for different durations up to 7 h.

**Battery tests**. In the case of NCM cathodes, a slurry consisting of NCM, super P, GB, and PVDF binder in a weight ratio of 97.0:0.5:1.0:1.5 was coated on an aluminum current collector and dried under vacuum. In the case of GB anode, a slurry consisting of GB and Li-polyacrylic acid (1.1 M) in a weight ratio of 95:5 was coated on a copper current collector and dried under vacuum. Both types of the electrode were then roll-pressed to enhance the interparticle contact. Coin-type half-cells (2032 size, Hoshen) were prepared by assembling the composite electrode, a separator (polypropylene, Celgard) and Li foil. 1.3 M lithium hexafluorophosphate ($LiPF_6$) in co-solvents of EC, EMC, and DEC in 3:4:3 = v:v:v with 1 wt% lithium bis(oxalate)borate (LiBOB) and 0.5 wt% vinyl ethylene carbonate (VEC) was used as electrolyte. For testing of cathodes in half-cell, constant current constant voltage (CCCV) mode was used for the 1st cycle at 0.475 mA with cutoff current being 0.0475 mA in the CV period. From the 2nd cycle, constant current (CC) mode was applied at various C-rates (1C = 4.75 mA). For testing of anodes in half-cell, CCCV mode was used for the 1st cycle at 0.27 mA with cutoff current being 0.027 mA in the CV period. From the 2nd cycle, CC mode was applied at various C-rates (1C = 2.7 mA). The mass loadings of NCM cathode and GB anode were $25.0 \pm 0.2$ and $3.8 \pm 0.2 \text{ mg cm}^{-2}$, corresponding to 4.75 and $2.7 \text{ mAh cm}^{-2}$, respectively.

The performance of the full-cells was evaluated in the voltage range of 2.8–4.2 V using the same electrolyte as in the half-cell tests. The n/p ratio, defined as the total capacity ratio between the anode and cathode, was 1.0, and the areal capacity was set to $2.4 \text{ mAh cm}^{-2}$. In the 1st cycle, CCCV mode was applied at 0.24 mA with cutoff current being 0.024 mA in the CV periods for both charging and discharging. From the 2nd cycle, CC mode was applied at various C-rates (1C = 2.4 mA). The mass loading of NCM for the full-cell was $12.6 \pm 0.2 \text{ mg cm}^{-2}$. The pre-lithiation of the GB anodes was conducted by applying $3.4 \text{ mA g}^{-1}$ in a half-cell containing a Li metal counter electrode until initial Coulombic efficiency reached 93.1%.

**Data availability**. All relevant data in this study are available from the corresponding authors upon request.

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

## Acknowledgements

This work was supported by funds from Samsung Electronics Co. Ltd.

## Author contributions

I.H.S. and J.W.C. designed the research. I.H.S. synthesized the samples. I.H.S., J.H.P., K.P., S.H., and J.S. carried out the physical and electrochemical characterizations. S.P. carried out the TEM analyses. I.H.S. and J.W.C. wrote the manuscript and supervised the project. S.-G.D., Y.H., and H.C. advised the project. All authors discussed the results and commented on the manuscript.

## Additional information

**Competing interests:** A relevant patent application is in progress by Samsung Electronics Co. Ltd. The remaining authors declare no competing financial interests.

