## [Peer Review File · Nature Communications]

Reviewers' comments:

Reviewer #1 (Remarks to the Author):

The authors have demonstrated graphene balls (GBs) obtained via chemical vapour deposition (CVD) growth to obtain high quality 3D graphene on SiO₂ nanoparticles within short time (30min). The GBs have shown significant improvement on the fast changing LIBs by mixing with commercial Lithium Nickel Cobalt Manganese Oxide (NCM) cathode material, and using itself alone as anode material. However, here are five points the authors have to clarify before the manuscript can be considered for publication.

1. This work uses a commercial NCM cathode material with a spherical structure shown in Fig3. Is there any reason why the author choice a commercial material with such particular shape? Is the GB has the same/similar effect when used in other cathode materials?
2. The authors have explained the remarkable effect of GBs on the cyclability of NCM in page 11-12, line 244-250. However, it is not clearly demonstrated how the cation can "accelerate the extraction of TM ions to the electrolyte, boosting the side reaction", neither by experimental results nor by literature. Electrochemical impedance spectroscopy and cyclic voltammetry measurements are needed to support this explanation.
3. The authors have attributed the "superior cycling performance of GB-NCM (line 257)" to the SiO₂ scavenge fatal HF as protection from side reaction in the electrolyte. However, if this process (SiO₂ scavenge HF) is the real reason for the enhanced performance, it will result in the generation of H₂O. This will determine stability issues, being also extremely dangerous for LIBs. How the authors remove the concern of safety if involve SiO₂ in the system?
4. The synthesis procedure is rather complicated and time consuming. The authors should comments on other more simple approaches for the realization of graphene-based anode (e.g. Nano Lett. 14, 4901, 2014). Moreover, the quality of the as-produced material is quite poor.
5. The irreversible capacities at the 1st charge-discharge cycles of each sample have to be presented or discussed for both cathode and anode, respectively. It is importance to evaluate the "real" performances of the active materials, especially in view of full battery integration.

Reviewer #2 (Remarks to the Author):

This manuscript reports the CVD growth of graphene on the catalytic active sites of silica NPs under CH₄ flow. The as-prepared graphene-silica assembly (i.e., graphene ball) was then uniformly coated on surface of commercial LiNi_{0.6}Co_{0.1}Mn_{0.3}O₂ (NCM) cathode material at a low weight content of 1wt% via Nobilta ball milling. The graphene ball (GB) coating improves the rate capability and cycle stability of NCM by increasing the electric conductivity of NCM and protecting the NCM from side reactions with electrolyte. A full cell with graphene-ball modified NCM as cathode and graphene-ball as anode was also assembled and showed an increased volumetric energy density of 27.6% compared to NCM//graphite cell. In spite of the systematic study and reasonable discussion in this manuscript, it lack of novelty to be warranted for publication in Nature communication. The submission to a more specialized journal is suggested. The details comments are as follows.

- (1) Why the specific surface area of as-prepared GB decreases with the increase of CVD reaction time (Table 1).
- (2) In Figure 4a, the authors reported that the charging capacity of GB-NCM at 5 and 10C are 144.6 and 105.3 mAhg⁻¹, respectively, which are the initial charge capacity at each current rate. However, the capacity clearly decreased to ~ 125 and 80 mAhg⁻¹, respectively, in the following cycles. Thus, it

is suggested to provide stable charge capacity values and/or average charge capacity values at each rate.

(3) In page 12, the authors claimed that "with a 4.5V cut-off, GB-NCM and pristine NCM preserved 97.3% and 92.1% of their original capacities after 100 cycles ". On the basis of rough calculation from Figure 4c, the initial capacity of GB-NCM with 4.5V cut-off is $\sim 210 \text{ mAh g}^{-1}$, and $\sim 190 \text{ mAh g}^{-1}$ at 100th cycle. Thus, the capacity retention is only 90.5%, which differs greatly with the reported 97.3%. So this discrepancy needs to be addressed.

(4) What is the significance of the reported CVD method as compared to other existing CVD growth approaches?

(5) How the capacity values of GB-NCM and GB compare with those obtained in other approaches?

Response to Reviewers' Comments

Reviewer #1

The authors have demonstrated graphene balls (GBs) obtained via chemical vapour deposition (CVD) growth to obtain high quality 3D graphene on SiO₂ nanoparticles within short time (30min). The GBs have shown significant improvement on the fast changing LIBs by mixing with commercial Lithium Nickel Cobalt Manganese Oxide (NCM) cathode material, and using itself alone as anode material. However, here are five points the authors have to clarify before the manuscript can be considered for publication.

Response: We are very thankful to the reviewer for his/her thorough review of our manuscript and constructive comments.

1. This work uses a commercial NCM cathode material with a spherical structure shown in Fig3. Is there any reason why the author choice a commercial material with such particular shape? Is the GB has the same/similar effect when used in other cathode materials?

Response: We intentionally chose the commercial spherical NCM particles because the current research pursued immediate commercial impact. It is well-known that most commercial cells adopt micron-sized spherical NCM particles that are synthesized by co-precipitation processes due to their high physical strengths and low surface areas (low side reactions).

In response to the reviewer's comment, we have examined the GB for an over-lithiated oxide (OLO in abbreviation, $\text{Li}_{1.18}\text{Ni}_{0.17}\text{Co}_{0.1}\text{Mn}_{0.56}\text{O}_2$). As shown below, although the cycling performance is not as compelling as that of NCM due to the intrinsic shortcomings of OLO (i.e., oxygen release, cation mixing, and etc.), GB-OLO exhibited enhanced cyclability compared to that of the pristine counterpart, implying the surface protection based on the GB coating is effective for other cathode materials besides NCM. We did not include the present results of the OLO electrodes in the manuscript because our manuscript targets commercial impact, and OLO electrodes are not generally as mature yet at present. We kindly ask the reviewer to understand this point.

OLO electrode condition:

OLO/ ketjen black (EC-300J)/PVDF binder=90/5/5, GB content in OLO=1 wt%

Cell configuration: half-cell with Li metal counter electrode

Operation condition: 4.7-2.5V, 25 °C, 0.1C @preycling, 1C @subsequent cycles, 1C=8 mA cm⁻²

1st cycle specific capacity: 227.7 mAh g⁻¹ (pristine OLO), 238.4 mAh g⁻¹ (GB-OLO)

1st cycle Coulombic efficiency: 82.2% (pristine OLO), 83.7% (GB-OLO)

Fig. cycling performance of pristine OLO and GB-OLO. Specific capacity (left) and capacity retention (right).

2. The authors have explained the remarkable effect of GBs on the cyclability of NCM in page 11-12, line 244-250. However, it is not clearly demonstrated how the cation can “accelerate the extraction of TM ions to the electrolyte, boosting the side reaction”, neither by experimental results nor by literature. Electrochemical impedance spectroscopy and cyclic voltammetry measurements are needed to support this explanation.

Response: We agree with the reviewer that it is valuable to further elucidate the effect of cation mixing on the interfacial stability. While cation mixing is a well-known mechanism for capacity degradation *via* metal dissolution (doi:10.3390/inorganics2010132, doi.org/10.1016/j.mattod.2014.10.040, doi.org/10.1016/j.mseb.2014.11.014, DOI: 10.1002/anie.201409262, and etc.), our metal dissolution test at 50 °C (Supplementary Table 3) indeed supports the effect of the GB coating; the dissolved amount of Ni decreased significantly by the GB coating, as compared with that of the pristine counterpart.

Separately, differential capacity (dQ/dV) profiles indicate smaller overpotentials for GB-NCM, in reflection of smaller interfacial resistance due to decreased metal dissolution. These measurements were performed at the same conditions as in Figure 4d (half-cell, 4.5 V cut-off, 60 °C). These data are also consistent with those in Supplementary Figure 7a in which GB-NCM exhibits lower overpotentials than those of pristine NCM.

Figure. (a) The differential capacity data (dQ/dV) extracted from the charge/discharge profiles of (b) pristine NCM and (c) GB-NCM half-cells: 4.5 V cut-off, 60 °C.

Following the reviewer’s suggestion, the interfacial resistance was additionally measured using electrochemical impedance spectroscopy (EIS) analysis (see below). Consistent with the above dQ/dV results, GB-NCM exhibited smaller semi-circles (or interfacial resistances) at the 1st and 50th cycle. These results are now included in Supplementary Figure 7b-c.

[Page 12]

...involving the electrolyte. The protective role of GB was reflected in the electrochemical impedance spectroscopy (EIS) analysis (Supplementary Fig. 7b-c).

Supplementary Figure 7 | The electrochemical properties of both NCM electrodes at 25 °C. a, 1st charge-discharge profiles at 0.1C and 1C. **(b-c)** Electrochemical impedance spectroscopy (EIS) plots after **b,** 1st cycle and **c,** 50th cycle.

Finally, metal dissolution was additionally characterized in a full-cell configuration (18650 type). In this experiment, two cells consisting of GB-NCM//Si+graphite and pristine NCM//Si+graphite were examined. After 300 cycles at 1C and 45 °C in the potential range of 4.3-2.8 V, transition metal (TM) concentrations at the anode surfaces were analyzed using EDX-SEM. As displayed below, the GB-

NCM//Si+graphite cell exhibited lower intensities with regard to TMs, F, and P. F and P are the indicators of SEI formation in a way that their high intensities represent amplified SEI growth.

Figure A. EDX-SEM elemental mapping on the anode surface of the pristine-NCM//Si+graphite cell after 300 cycles at 1C at 45 °C in the potential range of 4.3-2.8 V.

Figure B. EDX-SEM elemental mapping on the anode surface of the GB-NCM//Si+graphite cell after 300 cycles at 1C at 45 °C in the potential range of 4.3-2.8 V.

3. The authors have attributed the “superior cycling performance of GB-NCM (line 257)” to the SiO₂ scavenge fatal HF as protection from side reaction in the electrolyte. However, if this process (SiO₂ scavenge HF) is the real reason for the enhanced performance, it will result in the generation of H₂O. This will determine stability issues, being also extremely dangerous for LIBs. How the authors remove the concern of safety if involve SiO₂ in the system?

Response: We first agree with the reviewer that H₂O is problematic in LIBs by giving rise to corrosion reactions with electrode materials and safety hazards. It is reminded that the main reason for the performance degradation originating from the presence of water is the generation of HF by the reaction between H₂O and F-containing salt. HF can give rise to corrosion of cathode materials. Thus, HF and H₂O are in a reaction loop, and the scavenge of HF is net-effective in mitigating the side effects.

The effect of SiO₂ in scavenging HF and thus protecting electrode materials has been indeed well-adopted (references below) in the field; once SiO₂ reacts with HF, the reaction yields a partially fluorinated derivative more dominantly than H₂O. This reaction can also be understood as Lewis acid-base reaction such that the acidity of HF is neutralized by SiO₂, which is Lewis base. Also, the same effect was reported for Al₂O₃ and ZnO, both of which long have been used as protective coating materials of active materials for the same purpose.

References

- Sharabi, R., et al. "Significantly improved cycling performance of LiCoPO 4 cathodes." *Electrochem. Comm.* 13.8 (2011): 800-802.
- Tebbe, Jonathon L., Aaron M. Holder, and Charles B. Musgrave. "Mechanisms of LiCoO₂ cathode degradation by reaction with HF and protection by thin oxide coatings." *ACS appl. Mater. Inter.* 7.43 (2015): 24265-24278.
- Sclar, Hadar, et al. "The effect of ZnO and MgO coatings by a sono-chemical method, on the stability of LiMn_{1.5}Ni_{0.5}O₄ as a cathode material for 5 V Li-ion batteries." *J. Electrochem. Soc.* 159.3 (2012): A228-A237.
- Myung, Seung-Taek, et al. "Role of alumina coating on Li- Ni- Co- Mn- O particles as positive electrode material for lithium-ion batteries." *Chem. Mater.* 17.14 (2005): 3695-3704.

4. The synthesis procedure is rather complicated and time consuming. The authors should comment on other more simple approaches for the realization of graphene-based anode (e.g. Nano Lett. 14, 4901, 2014). Moreover, the quality of the as-produced material is quite poor.

Response: We thank the reviewer for bringing the reference to our attention. The given reference reports solution-processed graphene nanoflake ink to use graphene nanoflakes as an LIB anode. While solution processes have advantages for large-scale processing, the given graphene nanoflakes were produced by the exfoliation of graphite using sonication so that the quality of graphene is likely to be worse than our CVD-grown graphene, as indeed proven by the lower I_D/I_G of our GB (1.1 vs. 1.5) in the Raman analysis. In the quality control viewpoint, solution-processed graphene synthesis requires drying steps, during which the quality of graphene could vary from batch to batch. From the perspective of electrochemical performance, the SiO_x in our GB would deliver high volumetric capacities compared to those of graphene nanoflakes, as our GB contains high capacity SiO_x . More importantly, the main role of our GB with a hierarchical SiO_x -graphene structure is to be used as a coating material for NCM *via* an easy milling process to protect the NCM from side reactions. This hierarchical structure allows the uniform coating of the GB even at its small content (<2 wt%), which must be nontrivial with conventional solution-processed graphene family, including the nanoflakes in the referred paper, due to serious agglomeration.

We appreciate the reviewer's comment related to this reference and thus added the following sentence in the revised manuscript:

[Page 15]

...800 Wh L⁻¹ in a commercial cell setting, a remarkable milestone in the development of LIBs for next generation EVs. Graphene flakes produced by the solution-based exfoliation of graphite was previously integrated with LiFePO₄ cathode to build a full-cell³⁹. Despite the decent cyclability, the graphene nanoflake anode is expected to offer a smaller volumetric capacity than that of the GB containing high capacity SiO_x , leading to an inferior volumetric energy density of the corresponding full-cell.

5. The irreversible capacities at the 1st charge-discharge cycles of each sample have to be presented or discussed for both cathode and anode, respectively. It is importance to evaluate the “real” performances of the active materials, especially in view of full battery integration.

Response: As the reviewer pointed out, the first reversibility of both electrodes is important for the design and performance of practical full-cells. In response to the reviewer's comment, a description on how to assemble both electrodes into a full-cell is now added in the main text as follows:

[Page 14]

... The n/p ratio, defined by the capacity ratio between pre-lithiated anode and pristine cathode, was 1.0. In pairing these two electrodes in a full-cell, the following specifications of both electrodes were taken into consideration: whereas the specific capacities of the pre-lithiated GB and NCM are 716.2 mAh g⁻¹ and 190 mAh g⁻¹ at 0.1C, their ICEs are 99.8% and 93.7%, respectively. With this full-cell design, the first specific capacities of the anode and cathode in the actual cell were 666.6 mAh g⁻¹ and 178.2 mAh g⁻¹ at 25 °C, respectively, leading to an ICE of 93.1% (Supplementary

Fig. 10a and Table 5).

Supplementary Figure 10 | Charge-discharge profiles of the GB-NCM//GB full-cell. **a**, The first cycle at 0.1C. **b**, The second cycle at 25 °C and 60 °C when measured at 5C.

Reviewer #2

This manuscript reports the CVD growth of graphene on the catalytic active sites of silica NPs under CH₄ flow. The as-prepared graphene-silica assembly (i.e., graphene ball) was then uniformly coated on surface of commercial LiNi_{0.6}Co_{0.1}Mn_{0.3}O₂ (NCM) cathode material at a low weight content of 1wt% via Nobilta ball milling. The graphene ball (GB) coating improves the rate capability and cycle stability of NCM by increasing the electric conductivity of NCM and protecting the NCM from side reactions with electrolyte. A full cell with graphene-ball modified NCM as cathode and graphene-ball as anode was also assembled and showed an increased volumetric energy density of 27.6% compared to NCM//graphite cell. In spite of the systematic study and reasonable discussion in this manuscript, it lack of novelty to be warranted for publication in Nature communication. The submission to a more specialized journal is suggested. The details comments are as follows.

Response: We first thank the reviewer for his/her thorough review of our manuscript and constructive comments. As for the novelty, we would kindly emphasize that uniform coating of NCM (or LIB cathode materials in general) with a small content (i.e., <2 wt%) is nontrivial with existing graphene materials or other carbon nanomaterials due to their severe agglomeration. To overcome the agglomeration of graphene, we introduced the SiO_x-graphene assembly bearing a 3D hierarchical structure for the uniform coating even with ~1 wt% content. The given surface treatment turned out to be compatible with scalable Nobilta milling processes and also largely maintain the high energy density of the corresponding full-cell. We would kindly request the reviewer to view the current work focusing on the commercial impact of the GB coating in terms of improved cell performance and easy processing. Also, the efficient graphene growth based on a CVD process engaging OH⁻ as a mild oxidant is new and is expected to provide an insight to the graphene community.

(1) Why the specific surface area of as-prepared GB decreases with the increase of CVD reaction time (Table 1).

Response: We would appreciate the reviewer's careful review of our data. The decrease in specific surface area is attributed to the graphitization during which graphene layers are merged and stacked. The given trend in specific surface area is consistent with that of total pore volume (the column now added in Table 1, see below). Also, the progression of graphitization was reflected in the grown particle sizes; in revised Supplementary Fig. 3; the SEM images taken after 60 min and 240 min are added:

[Page 7]

... specific surface area, and conductivity of GB at different CVD growth time are summarized in Table 1. As for specific surface area, its decrease with increasing reaction time is attributed to the progressive graphitization during which graphene layers are merged and stacked...

Table 1 | Various properties of graphene-ball at different CVD reaction time.

Time	Graphene content in	D/G peak	Specific surface area	Total pore volume	Conductivity (S cm ⁻¹)
------	---------------------	----------	-----------------------	-------------------	------------------------------------

	mass	ratio	$(\text{m}^2 \text{g}^{-1})$	$(\text{cm}^3 \text{g}^{-1})$	
0 min	-	-	175.5	0.5821	$<10^{-7}$
5 min	17.1%	1.102	147.1	0.5535	5.8
30 min	42.5%	1.076	108.3	0.2875	22.6
60 min	56.2%	1.066	81.9	0.2339	48.4
240 min	84.5%	1.055	35.6	0.1978	65.5

Supplementary Figure 3 | Morphology and color of GB at different CVD growth time. (a-c) SEM images of GB after a, 0 min, b, 5 min, c, 30 min, d, 60 min, and e, 240 min.

(2) In Figure 4a, the authors reported that the charging capacity of GB-NCM at 5 and 10C are 144.6 and 105.3 mAhg⁻¹, respectively, which are the initial charge capacity at each current rate. However, the capacity clearly decreased to ~ 125 and 80 mAhg⁻¹, respectively, in the following cycles. Thus, it is suggested to provide stable charge capacity values and/or average charge capacity values at each rate.

Response: Once again, we appreciate the reviewer's careful review. In response to the reviewer's comment, we replaced those values with averaged ones at both C-rates. Consistently, the values at other C-rates were also corrected with averaged values:

[Page 11]

... . As the charging rate was raised from 0.1C to 0.2C, 0.5C, 1C, 2C, 5C, and 10C, the charging capacity of GB-NCM decreased from 191.6 mAh g⁻¹ to 188.4 mAh g⁻¹, 183.5 mAh g⁻¹, 175.3 mAh g⁻¹, 161.6 mAh g⁻¹, 131.2 mAh g⁻¹, and 86.1 mAh g⁻¹, respectively (Fig. 4a). In contrast, with the same increases in C-rate, the charging

capacity of pristine NCM dropped more severely, from 187.2 mAh g⁻¹ to 178.3 mAh g⁻¹, 160.5 mAh g⁻¹, 133.4 mAh g⁻¹, 101.8 mAh g⁻¹, 54.2 mAh g⁻¹, and 1.3 mAh g⁻¹, respectively. When tested at 60 °C (Fig. 4b), the gap between both electrodes became more narrow, presumably due to the increased electronic conductivity of the NCM. However, the superior charging capability of GB-NCM was preserved; for example, while both electrodes exhibited almost the same charging capacities at 0.1C, GB-NCM showed 31.4% (171.7 mAh g⁻¹ vs. 130.7 mAh g⁻¹) and 126.7% (134.9 mAh g⁻¹ vs. 59.5 mAh g⁻¹) higher capacities at 2C and 10C, respectively.

(3) In page 12, the authors claimed that "with a 4.5V cut-off, GB-NCM and pristine NCM preserved 97.3% and 92.1% of their original capacities after 100 cycles ". On the basis of rough calculation from Figure 4c, the initial capacity of GB-NCM with 4.5V cut-off is ~210 mAh g⁻¹, and ~190 mAh g⁻¹ at 100th cycle. Thus, the capacity retention is only 90.5%, which differs greatly with the reported 97.3%. So this discrepancy needs to be addressed.

Response: In fact, the first cycles in Fig. 4c and d with all cut-off conditions were measured at 0.1C, whereas the subsequent cycles were measured at 1C. Hence, the capacity retentions were evaluated with respect to the capacities at the second cycles. To clarify, the text and Supplementary Table 2 are now revised as follows:

[Page 12]

...For example, when tested for 100 cycles at 1C with a 4.5 V cut-off, GB-NCM and pristine NCM preserved 97.3% and 92.1% of the capacities in the second cycles, respectively. The first cycles with all cut-off voltages were measured at 0.1C. This superior cyclability was also retained at 60 °C (Fig. 4d), which is a critical temperature for EV operations. With the same 4.5 V cut-off, after 50 cycles, the capacity retentions of GB-NCM and pristine NCM were changed to 91.0% and 79.9% with respect to their capacities in the second cycles, respectively.

[Supplementary Table 2]

Supplementary Table 2 | Summary of electrochemical properties of LiNi_{0.6}Co_{0.1}Mn_{0.3}O₂ (NCM) in half-cell measurements at 25 °C and 60 °C.

25 °C		Capacity @ 0.1C (mAh g _{NCM} ⁻¹)	initial Coulombic efficiency (%)	Average Coulombic efficiency for 2-100 cycles (%)	Capacity retention for 2-100 cycles (%)
Cut off 4.3 V	Pristine NCM	189.5	93.15	99.92	97.8
	GB-NCM	192.2	93.71	99.96	99.5 Δ1.7%↑
Cut off 4.4 V	Pristine NCM	198.5	92.06	99.78	95.0
	GB-NCM	201.1	92.32	99.93	98.5 Δ3.5%↑
Cut off 4.5 V	Pristine NCM	210.3	91.95	99.63	92.1
	GB-NCM	211.5	92.89	99.90	97.3 Δ5.2%↑
60 °C		Capacity @ 0.1C (mAh g _{NCM} ⁻¹)	initial Coulombic efficiency (%)	Average Coulombic efficiency for 2-50 cycles (%)	Capacity retention for 2-50 cycles (%)

Cut off 4.3 V	Pristine NCM	184.5	96.10	99.64	93.22	
	GB-NCM	191.4	96.62	99.68	97.10	$\Delta 4.16\uparrow$
Cut off 4.4 V	Pristine NCM	198.9	91.92	99.58	92.70	
	GB-NCM	206.5	95.71	99.62	95.07	$\Delta 2.56\uparrow$
Cut off 4.5 V	Pristine NCM	209.5	89.54	99.33	79.91	
	GB-NCM	215.1	91.96	99.46	90.99	$\Delta 13.9\uparrow$

(4) What is the significance of the reported CVD method as compared to other existing CVD growth approaches?

Response: As described in page 5~7, the significance of the current CVD process as compared with previous CVD processes that rely on metal catalysts can be summarized as follows:

1. The current report is the first demonstration of a 3D hierarchical structure in which high quality graphene grows from SiO_x centers. Thus, graphene layers can be coated onto the battery cathode material via a Nobilta milling process while the graphene content is minimal (i.e., <2 wt%, thus minimal sacrifice in energy density). The graphene grown from CVD processes relying on metal catalysts cannot be integrated at such small contents due to agglomeration.
2. The use of SiO_x is also unique and critical because SiO_x serves as a catalyst for graphene growth. Moreover, the generated OH^- by the reaction between SiO_2 and CH_4 serves as a mild oxidant, without which the growth of amorphous carbon becomes significant, ruining the quality of the carbon product. The present work is the first demonstration of utilizing SiO_x and OH^- with the corresponding roles (catalyst and mild oxidant, respectively) in graphene growth.
3. Finally, the current CVD scheme does not include H_2 in the reaction, unlike typical cases employing metal catalysts. This means that the choice of reactants is adjusted for efficient growth of graphene from SiO_2 particles. The less efficient graphene growth of the reaction including H_2 was demonstrated in Supplementary Fig. 2.

Based on the reviewer's comment, we feel that we would better add a sentence as follows to clearly deliver the distinct features of our CVD reaction scheme as compared with other previous ones:

[Page 6]

...revealing the importance of the exclusive use of CH_4 on the efficient graphene growth in our synthesis protocol. Overall, the choice of reactants and reaction temperature in the present reaction scheme is distinct from previous CVD-based counterparts employing metal catalysts and plays a key role in producing a graphene- SiO_x assembly with the 3D hierarchical structure.

(5) How the capacity values of GB-NCM and GB compare with those obtained in other approaches?

Response: In response to the reviewer’s comment, we conducted a literature survey with respect to specific capacity.

For the cathode, we limited the material scope to the current NCM613 composition, as it is well-known that Ni-content mainly determines the specific capacity. The comparison with other high Ni compositions is too broad to set a boundary. The choice of the 613 composition was based on our internal research in the past five years or so, focusing on its immediate commercial impact by taking various properties (full-cell energy density, cycle life, tap density, price, etc.) into consideration. The given choice is also being intensively investigated by other battery makers targeting commercial application in the near future.

For the anode, we compared the specific capacity of the GB with those of reported graphene-Si composites (refs 4 – 8) as well as solely graphene (refs 9, 10).

As for the cathode, while the initial gravimetric capacities are similar due to the fixed composition, the capacity of the GB-NCM after 100 cycles was clearly higher than those in previous literatures because of the various beneficial effects of the GB coating (mitigation of side reactions, HF scavenging, etc.).

As for the anode, an accurate and fair comparison appears difficult because cells were prepared and measured in different conditions (mass loading, C-rate, etc.). In particular, the areal capacity of the GB electrode (2.7 mAh cm^{-2}) falls in the range of commercial LIBs and is higher than those of most electrodes in the references listed below. Unfortunately, many of the given references do not provide information regarding areal capacity (perhaps indicating different electrode loadings), so a direct comparison is infeasible. More importantly, the specific capacity of the anode is indeed high enough to achieve a substantial increase in energy density; together with the increased content of NCM in the cathode (92 \rightarrow 97 wt%) by integrating the GB coating, the replacement of graphite with GB increases the energy density by 33.5% (see details in Supplementary Table 1).

We feel that this comparison is not appropriate to include in the manuscript because the measurement/electrode conditions are not consistent. We hope that the reviewer understands this point.

	Capacity (mAh g ⁻¹)	C-rate	Voltage window	Cycle
Ref [1]	175	0.5C	2.5 – 4.4 V	100 th
Ref [2]	182	0.5C	2.5 – 4.4 V	100 th
Ref [3]	145	0.5C	3.0 – 4.3 V	100 th
This work	198	1C	2.5 – 4.4 V	100th

	Capacity (mAh g ⁻¹)	C-rate (or current density)	Voltage window	Cycle
Graphene-Si, Ref [4]	1720	100 mA g ⁻¹	0.05 – 1.0 V	1 st
Graphene-Si, Ref [5]	1100	1000 mA g ⁻¹	0.02 – 2.0 V	1 st
Graphene-Si, Ref [6]	3200	1000 mA g ⁻¹	0.02 – 1.5 V	1 st
Graphene-Si, Ref [7]	2300	120 mA g ⁻¹	0.01 – 2.0 V	1 st
Graphene-Si, Ref [8]	1000	500 mA g ⁻¹	0.01 – 1.5 V	1 st
Graphene only, Ref [9]	650	1C	0.03 – 3.0 V	1 st
Graphene only, Ref [10]	1264	100 mA g ⁻¹	0.01 – 3.5 V	1 st
This work	716.2	70 mA g⁻¹	0.01 – 1.5 V	1st

Cathode

- [1] Chem. Mater. 2015, 27, 7370–7379
[2] Electrochimica Acta 2017, 230, 308–315
[3] Electrochimica Acta 2017, *accepted* (DOI: <http://dx.doi.org/doi:10.1016/j.electacta.2017.05.148>)

Anode

- [4] Adv. Energy Mater. 2012, 2, 1086–1090
[5] J. Phys. Chem. Lett. 2012, 3, 1824–1829
[6] Adv. Energy Mater. 2011, 1, 1079–1084
[7] Adv. Mater. 2014, 26, 758–764
[8] Small 2013, 9, 16, 2810–2816
[9] Carbon 2009, 47, 2049–2053
[10] Electrochimica Acta 2010, 55, 3909–3914

Reviewers' comments:

Reviewer #1 (Remarks to the Author):

The authors have fully answered to all the reviewers' comments/doubts. In view of this referee, seen the content and the novelty, the manuscript can now be considered for publication.

Reviewer #3 (Remarks to the Author):

In this manuscript, the authors describe a chemical vapor deposition process to grow a graphene-silica 3D assembly, with the SiO_x nanoparticle center allows 1 wt% graphene-ball to be uniformly coated onto a nickel-rich layered cathode via mild Nobilta milling. The authors claim that graphene-ball coating improves cycle life and fast charging capability by protecting the electrode surface from detrimental side reactions and providing efficient conductive pathways. This is a well-organized manuscript and exhibit thoughtful consideration about the coating effect of GB using various analysis method. Although the synthetic method employed in the manuscript does not exhibit significant novelty compared to other previous works, the uniform coating of 1 wt% GB onto NCM surface using an industrial scalable process is impressive. The authors also provide very comprehensive explanations to address the questions from the previous reviewers. Following additional comments should be considered by the authors to further improve the manuscript:

1. In the supplementary Table 1 "Design parameters for an EV prismatic cell", when comparing the NCM//graphite and the GB-NCM//GB, the authors should provide explanation of why the NP ratio was changed from 1.08 to 1.0?
2. When comparing the energy density of any cells, the authors should make it clear which component is included and which one is not. For example, in supplementary Table 1: energy density of 947.1 Wh/L was mentioned for NCM//graphite and 1300.4 Wh/L for GB-NCM//GB. However, on page 15, line 332 and 333, energy density of 630 Wh/L and 800 Wh/L were mentioned for the same chemistry.
3. On page 15, the authors claim that the superior performance of the GB-NCM//GB full-cell is due to the critical role of GB protection on the NCM. If this is true, was the similar performance observed in a GB-NCM//graphite full-cell?

Response to Reviewers' comments

Reviewer #1 (Remarks to the Author):

The authors have fully answered to all the reviewers' comments/doubts. In view of this referee, seen the content and the novelty, the manuscript can now be considered for publication.

Response: We are very pleased to hear this positive evaluation.

Reviewer #3 (Remarks to the Author):

In this manuscript, the authors describe a chemical vapor deposition process to grow a graphene-silica 3D assembly, with the SiO_x nanoparticle center allows 1 wt% graphene-ball to be uniformly coated onto a nickel-rich layered cathode via mild Nobilta milling. The authors claim that graphene-ball coating improves cycle life and fast charging capability by protecting the electrode surface from detrimental side reactions and providing efficient conductive pathways. This is a well-organized manuscript and exhibit thoughtful consideration about the coating effect of GB using various analysis method. Although the synthetic method employed in the manuscript does not exhibit significant novelty significant novelty compared to other previous works, the uniform coating of 1 wt% GB onto NCM surface using an industrial scalable process is impressive. The authors also provide very comprehensive explanations to address the questions from the previous reviewers. Following additional comments should be considered by the authors to further improve the manuscript:

Response: We are very grateful to the reviewer for his/her thorough review and thoughtful comments.

1. In the supplementary Table 1 "Design parameters for an EV prismatic cell", when comparing the NCM//graphite and the GB-NCM//GB, the authors should provide explanation of why the NP ratio was changed from 1.08 to 1.0?

Response: We appreciate the reviewer's careful review of our data. The different NP ratios originate from the different initial Coulombic efficiencies (ICEs) of the electrode materials in both cells. In the NCM//graphite cell, the ICEs of the NCM and graphite are ~93% and ~95%, respectively. With these ICE values, NP ratio of around 1.1 (1.08 in our case) is widely adopted to have an excessive capacity in the anode and thus prevent Li deposition on the anode surface. By contrast, in the GB-NCM//GB cell, pre-lithiated GB has a higher ICE of 99% so that NP ratio was decreased to 1.0. This modification is based on the fact that the increased

reversibility of the anode loses a smaller amount of Li ions during the 1st cycle. This explanation is now added as follows:

[Supplementary Table 1]

Parameter		NCM//graphite	GB-NCM//GB
EV Cell	Current density (mAh cm ⁻²)	4.4	4.4
	NP ratio	1.08 ^a	1.0 ^a
	Packing ratio (%)	90.0	90.0
	Energy Density (Wh L ⁻¹)	630	800
	Total capacity (Ah@0.33C)	49.67	63.40 ($\Delta 27.6\% \uparrow$)

^aThe different NP ratios originate from different initial Coulombic efficiencies (ICEs) of electrode materials. In the NCM//graphite cell, the ICEs of the NCM and graphite are ~93% and ~95%, respectively. With these ICE values, NP ratio of around 1.1 is widely adopted to have an excessive capacity in the anode. In the GB-NCM//GB cell, pre-lithiated GB has a higher ICE of 99% so that NP ratio was decreased to 1.0.

2. When comparing the energy density of any cells, the authors should make it clear which component is included and which one is not. For example, in supplementary Table 1: energy density of 947.1 Wh/L was mentioned for NCM//graphite and 1300.4 Wh/L for GB-NCM//GB. However, on page 15, line 332 and 333, energy density of 630 Wh/L and 800 Wh/L were mentioned for the same chemistry.

Response: We fully agree with the reviewer that the conditions for energy density estimation would better be described in detail. The energy densities in the bottom table of Supplementary Figure 1 were based on the mass of active materials only. By contrast, the energy densities on page 15 were obtained targeting an EV cell by taking material packing, electrode density, and content of active material into consideration. Thus, the latter estimation is more realistic. To avoid similar confusion, the energy densities of EV cells were added in the top table, and the title of the bottom table was revised to “**Material** Energy Density = $(C_{\text{cathode}} \times C_{\text{anode}} \times V_{\text{nominal}}) / (C_{\text{cathode}} + C_{\text{anode}})$ ” as follows:

[Supplementary Table 1]

Parameter	NCM//graphite	GB-NCM//GB
-----------	---------------	------------

EV Cell	Current density (mAh cm ⁻²)	4.4	4.4
	NP ratio	1.08	1.0
	Packing ratio (%)	90.0	90.0
	Energy Density (Wh L ⁻¹)	630	800
	Total capacity (Ah@0.33C)	49.67	63.40 (Δ27.6%↑)

Material Energy Density = $(C_{\text{cathode}} \times C_{\text{anode}} \times V_{\text{nominal}}) / (C_{\text{cathode}} + C_{\text{anode}})$

Cell type	C _{anode}		C _{cathode}		V _{nominal}	Energy density	
	[mAh g ⁻¹]	[mAh cc ⁻¹]	[mAh g ⁻¹]	[mAh cc ⁻¹]	V	[Wh kg ⁻¹]	[Wh L ⁻¹]
pristine NCM//graphite	360	576	190	570	3.4	422.8	974.1
GB-NCM//GB	700	770	190	760	3.4	508.1 (Δ20.2%↑)	1300.4 (Δ33.5%↑)

3. On page 15, the authors claim that the superior performance of the GB-NCM//GB full-cell is due to the critical role of GB protection on the NCM. If this is true, was the similar performance observed in a GB-NCM//graphite full-cell?

Response: Yes, indeed. As shown below, GB-NCM//graphite and pristine NCM//graphite cells were tested at 25 and 60 °C. At both temperatures, GB-NCM//graphite cell showed better performance in specific capacity and cycle life, reconfirming the beneficial effect of GB as a cathode coating material. This phenomenon is attributed to the fact that GB protects the NCM cathode from detrimental side reactions and gas evolution. For reference, once the areal capacity is increased to 2.4 mAh cm⁻² as in Figure 5 in the main text, the graphite anode does not operate as well as GB, because the graphite anode cannot afford to accommodate Li ions at this high C-rate (5C) during charge. This point is now reflected as follows:

[Page 16]

... a dramatic effect on the cycling performance and fast charging capability even in such challenging commercial cell conditions. The effect of GB was also demonstrated for full-cells incorporating graphite anodes (Supplementary Figure 11). Consistent with the half-cell results, the superior performance of GB-NCM//graphite full-cell as compared with that of

pristine NCM//graphite counterpart is attributed to the GB's protection of NCM against side reactions and gas evolution.

Supplementary Figure 11 The cycling performance and fast charging capability of NCM//graphite full-cells. (a-b) Charge-discharge profiles at a, 25 °C and b, 60 °C when measured at 5C ($1C=190 \text{ mAh g}_{\text{NCM}}^{-1}$). c, Capacity retentions at 5C. The areal capacities of all the cells in this figure are 1 mAh cm^{-2} . In each cycle, charging and discharging rates were the same for all the data shown in this figure. Note that once the areal capacity is increased to 2.4 mAh cm^{-2} as in Figure 5 in the main text, the graphite anode does not operate as well as GB, because the graphite anode cannot afford to accommodate Li ions at this high C-rate during charge.

REVIEWERS' COMMENTS:

Reviewer #3 (Remarks to the Author):

My concerns have been well addressed in the revised manuscript and the response letter. I recommend this revised manuscript for acceptance in its present form.